# Type 1 diabetes mellitus patients had lower total vitamin K levels and increased sensitivity to direct anticoagulants

Alejandro Carazo[1], Markéta Paclíková[2], Jaka Fadraersada[1], Raúl Alva-Gallegos[1], Pavel Skořepa[2,3], Catherine Gunaseelan[1], Kateřina Matoušová[4], Kristýna Mrštná[5], Lenka Kujovská Krčmová[4,5], Alena Šmahelová[2], Vladimír Blaha[2], Přemysl Mladěnka[1]*

1 Department of Pharmacology and Toxicology, Faculty of Pharmacy in Hradec Králové, Charles University, Hradec Králové, Czech Republic, 2 3rd Department of Internal Medicine-Metabolic Care and Gerontology, University Hospital and Faculty of Medicine in Hradec Králové, Charles University, Hradec Králové, Czech Republic, 3 Department of Military Internal Medicine and Military Hygiene, Military Faculty of Medicine, University of Defence, Hradec Králové, Czech Republic, 4 Department of Clinical Biochemistry and Diagnostics, University Hospital, Hradec Králové, Czech Republic, 5 Department of Analytical Chemistry, Faculty of Pharmacy in Hradec Králové, Charles University, Hradec Králové, Czech Republic

* mladenkap@faf.cuni.cz

## Abstract

The abnormalities in blood coagulation in patients with diabetes can lead to a pro-thrombotic state and requirement for the administration of direct anticoagulants. However, no comparative studies have been conducted on the effects of different direct anticoagulants. A head-to-head investigation of the impact of anticoagulants in 50 patients of type 1 diabetes mellitus (DMT1) was performed, and the data were compared to 50 generally healthy individuals. Prothrombin time (PT) and activated partial thromboplastin time (aPTT) were measured in plasma treated with vehicle, heparin, or four direct anticoagulants at 1 μM. In addition to common biochemical parameters, novel inflammatory markers (neopterin, kynurenine/tryptophan ratio) and major vitamin K forms were measured. Heparin and dabigatran treatments resulted in prolonged coagulation in DMT1 patients compared to healthy individuals in both tests (both p < 0.001). The same phenomenon was observed for rivaroxaban and apixaban-treated samples in PT (p < 0.001). Interestingly, healthy volunteers had higher total vitamin K levels than DMT1. Further analysis suggested that observed coagulation differences were not caused by differences in glycemia but were rather associated with an unexpected, better lipid profile of our DMT1 group. There were also correlations between prolongation of coagulation brought about by the most active anticoagulants and inflammatory markers, and hence inflammatory state probably also contributed to the differences, as well as the mentioned differences in vitamin K levels. Conclusively, this paper suggests the suitability for controlling the effects of direct anticoagulants in DMT1 patients.

**Data availability statement:** Anonymized research data are available at ZENODO (10.5281/zenodo.15148343).

**Funding:** This work was supported by the Czech Health Research Council (NU21J-02-00021); the project New Technologies for Translational Research in Pharmaceutical Sciences /NETPHARM, project ID CZ.02.01.01/00/22_008/0004607, co-funded by the European Union/, Charles University (SVV 260 549) and MH CZ - DRO (grant number UHHK, 00179906).

**Competing interests:** The authors have declared that no competing interests exist.

## Introduction

Coagulation is a critical physiological mechanism that prevents excessive bleeding while ensuring the repair of damaged blood vessels. This intricate cascade of events involves the orchestration of various factors, enzymes, and platelets [1]. An optimal balance between procoagulant and anticoagulant components is essential to maintain vascular integrity [2]. In healthy individuals, coagulation is indeed a tightly regulated process that ensures rapid clot formation at the site of injury, while simultaneously preventing uncontrolled clotting in the circulation [3]. However, disturbances in this delicate equilibrium are observed in several diseases. In particular, individuals with type 1 diabetes mellitus (DMT1) may experience alterations in the coagulation cascade due to multiple factors, including hyperglycemia, insulin deficiency, and chronic inflammation [4–6]. Studies have suggested that these alterations could contribute to an increased risk of thrombotic events, such as acute myocardial infarction and stroke, in these patients [7,8]. To prevent such cardiovascular events, the use of hypolipidemic, antiplatelet, and/or antihypertensive drugs is considered crucial [9]. Additionally, anticoagulant therapy, currently mainly consisting of direct oral anticoagulants (DOACs), is used under several situations encompassing atrial fibrillation [10]. Based on the aforementioned, in particular diabetic persons might profit from such therapy. There are currently no papers comparing the potency of direct anticoagulants in diabetic patients and non-diabetic persons. Based on the known differences, e.g., inflammatory state, which is known to be linked with coagulation disturbance as was well observed in COVID pandemic [11], we speculate that in addition to reported differences in coagulation [5], there might be differences in the effect of clinically used anticoagulants between diabetic and non-diabetic population. Moreover, a recent study in rats suggested a protective role of the direct thrombin inhibitor argatroban on cardiac dysfunction in a model of DMT1 [12].

Coagulation assays such as prothrombin time (PT) and activated partial thromboplastin time (aPTT) are frequently employed in clinical practice to determine the functionality of the processes and can be used for a simple assessment of the adequacy of the anticoagulant therapy. PT is a single-stage screening test used to evaluate extrinsic and common coagulation pathways, and thus shorter values are observed in hypercoagulability states. This method analyses mainly the activity of coagulation factors I, II, V, VII, or X [13]. Since PT test values can be affected by reagents used (manufacturer, different batches, etc.), an international normalized ratio (INR) is usually employed to standardize PT results to reliably compare results among laboratories [14]. On the other hand, aPTT is employed to evaluate the intrinsic and common pathways of the coagulation cascade. Importantly, vitamin K is needed for posttranslational modification of four coagulation proteins as well as three anticoagulant factors and hence it is an important player in functional hemostasis [15]. Dysfunction, deficiency, or inhibition in one of the following factors I, II, V, VII-XII leads to prolongation of coagulation [16]. There are data that prolonged PT is associated with increased mortality in coronary artery disease patients [17] and prolonged aPTT after a thrombin inhibitor treatment is linked with an increased risk of death, myocardial infarction, or refractory angina [18].

In this study, we aimed primarily to 1) analyse the coagulation response *via* the PT and aPTT assays in blood samples of 50 DMT1 patients and compare it to 50 age-matched generally healthy individuals, to 2) assess the impact of four DOACs (two factor Xa inhibitors rivaroxaban and apixaban as well as two direct thrombin inhibitors dabigatran and argatroban) at an equimolar concentration of 1 μM, which is an achievable plasma level of these anticoagulants in treated patients [19–21], to 3) see possible differences in vitamin K serum levels as well as their impact on coagulation and the effects of direct anticoagulants, and to 4) define possible other factors responsible for difference in coagulation responses to direct anticoagulants between DMT1 patients and healthy controls. For gaining additional insight, the dependence of coagulation parameters on hyperglycemia, high lactate concentration and hyperosmolarity, which can be found in untreated or poorly treated patients with diabetes, was investigated.

## Materials and methods

### Chemicals and drugs used in the study

Dimethyl sulfoxide (DMSO), dabigatran, and rivaroxaban were purchased from Sigma (St. Louis, MO, USA), saline solution 0.9% was bought from B. Braun (Melsungen, Germany), and heparin was acquired from Zentiva (Prague, Czech Republic). Apixaban was purchased from Toronto Research Chemical Inc. (Ontario, Canada), whereas argatroban from Eubio (Vienna, Austria).

### Donor and patient enrolment

A total of 50 generally healthy volunteers and 50 DMT1 patients were recruited at the University Hospital of Hradec Králové. The diagnosis for DMT1 was based on standard criteria [22]. Exclusion criteria for healthy subjects have been specified previously [23]. The same exclusion criteria were also applied for DMT1 patients. In general, serious illnesses (presence of tumors, heart failure, autoimmune diseases), abuse of alcohol, regular administration of antiplatelet and/or anticoagulant drugs were the exclusion criteria for both groups. Contrarily, given the impracticability of having completely healthy donors also in higher age categories, we enabled to include in the study also well-treated patients with hypertension and hypothyroidism. All participants signed informed consent and this study was approved by the Ethical Committees from both the University Hospital of Hradec Králové (No. 201907 S04P from June 21, 2019 and extended by No. 202007 I04 from June 30, 2020) and the Faculty of Pharmacy, Charles University (No. UKFaF/92240/2021–2 from March 3, 2021). Our experiments were performed in accordance with the Declaration of Helsinki. Blood samples were collected between the 10th of May 2021 and the 10th of March 2023.

### Blood collection and treatment

Blood collection was always performed in the morning between 8–9 a.m., and samples were delivered to our laboratory within 30 min. Donors and patients were informed not to consume alcohol or drugs affecting haemostasis (particularly non-steroidal anti-inflammatory drugs, and glucocorticoids) at least 24 hours before blood collection. Blood samples were collected by venipuncture into plastic disposable syringes containing citrate sodium (3.2%; BD Vacutainer, NJ, USA; for coagulation experiments) or a clotting accelerator (for biochemical assessment of basic metabolic parameters and vitamin K in serum). Citrated blood was carefully transferred to the laboratory for analysis under standard conditions and immediately centrifuged at 2,000 g for 20 minutes at room temperature to obtain (platelet-poor) plasma. Coagulation experiments were initiated within 1 hour after blood draw.

### Coagulation assays

PT and aPTT were determined using a four-channel semi-automated coagulometer Ceveron® (Technoclone, Vienna, Austria). Reagents needed for these experiments were purchased from the same company. DMSO in a final concentration

of 1% was included in the vehicle (negative) control in all experiments, and heparin was employed as the positive control (final concentration of 5 IU/mL for PT assay and 0.5 IU/mL for aPTT assay). The final concentration of all direct anticoagulants was 1 µM. Each condition was measured at least twice for each patient sample. PT values were expressed as INR, which was calculated in the following way: INR = (PT patient/PT normal)$^{ISI}$, where the international sensitivity index (ISI) is used as a normalizing factor. On the other hand, aPTT values were expressed as time in seconds.

## Impact of glycemia, lactate, and hyperosmolarity on coagulation

The determination of the effect of glycemia, lactate, and hyperosmolarity on PT (INR) and aPTT was performed as well with the above-mentioned coagulometer. Additionally, thrombin time (TT) was measured: plasma (200 µL) with solvent/compound was initially incubated for 1 min at 37 °C. After the incubation period, 200 µL of 3.3 IU of thrombin reagent (containing bovine thrombin, Technoclone) was added, and TT was recorded. Plasma samples for all tests (PT, aPTT, TT) were treated with saline (solvent, as the negative control), glucose (to achieve final concentration: 10–30 mM), lactate (1–10 mM), or NaCl (for hyperosmolarity simulation; an increase in Na$^+$ concentration of 20 mM). Glucose in plasma samples was measured before the experiment by the glucometer Wellion (Med Trust, Austria). In a separate set of experiments, plasma was incubated with glucose (at a final concentration of 30 mM) for 60 minutes at 37°C. All experiments were performed in at least in technical duplicates for each condition and in blood from 3 different healthy blood donors.

## Biochemical assays

Serum and urinary samples were protected against light and transported to the laboratory immediately after collection. Biochemical markers (glucose, serum, creatinine, neopterin, tryptophan, kynurenine, LDL, HDL, total cholesterol, triglycerides, and non-HDL) were measured in the serum of all blood samples. Additionally, creatinine, neopterin, tryptophan and kynurenine were measured as well in urine. Common markers were measured using commercial enzymatic kits by Cobas® and detected by the Cobas 8000 system (Roche Diagnostic, Basel, Switzerland). Glycated hemoglobin (HbA1c) was determined using a fully automated Arkray Adams HA-8180 (Arkray Inc., Japan). The principle was ion-exchange HPLC with gradient elution and VIS detection. Vitamin K was determined by the LC-MS/MS method [25]. The chromatographic separation was carried out on a Kinetex™ C18 column using a mobile phase consisting mainly of methanol. The target analytes were detected by electrospray ionization mass spectrometry. The simple sample preparation technique based on miniaturized liquid-liquid extraction was used for serum samples. Serum neopterin, kynurenine, and tryptophan were measured using HPLC-FLD/PDA method [24] while urinary neopterin, kynurenine, tryptophan and creatinine were determined using HPLC-FLD/PDA (a stationary phase consisting of two connected RP-18e monolithic columns 4.6 mm × 50 mm, 3.0 mm × 100 mm in combination with a 15 mM phosphate buffer as the mobile phase) [26] and UHPLC-MSMS method (the stationary phase Kinetex Polar C18 100 × 4.6 mm, 2.6 µm protected with security guard column EVO C18 3 mm ID). The mobile phase was composed of 65% 5 mM ammonium formate buffer and 35% methanol with 0.2% formic acid with a flow rate of 0.6 mL/min. Urinary creatinine was employed to correct for urine dilution.

## Statistical analysis

The sample size was estimated by a power analysis: to prove a difference of 10% in anticoagulant response (power analysis: p = 0.05, power 0.95) 26 patients in each group are needed. We have included 50 donors in both groups in order to be able to perform some subanalyses due to expected drop-outs. Coagulation data were compared using the Student's t-test or the Mann-Whitney test depending on the result of the normality test (the Shapiro-Wilk test). Correlation analyses were also performed based on the normality; in case of normal data distribution, the Pearson test was used; otherwise, the Spearman test was employed. All statistical studies were performed using GraphPad version. 10.1.2 software (GraphPad Software, San Diego, CA, USA).

 

# Results

## Characterization of DMT1 patients and generally healthy donors

In total, 50 participants of each group were recruited. Both groups were sex-balanced and age-matched. All recruited individuals fulfilled the inclusion and exclusion criteria, including absence of treatment with anticoagulant and antiplatelet drugs. However, several DMT1 patients were diagnosed with minor concomitant illnesses: 13 (26%) were diagnosed with hypertension and 6 (12%) with hypothyroidism. Generally healthy subjects were also diagnosed with hypertension in 9 cases (18%), and 6 volunteers (12%) had hypothyroidism. Two DMT1 patients and one generally healthy individual suffered both hypertension and hypothyroidism. As expected, DMT1 patients had significantly higher fasting glucose levels compared to those of the healthy group (Table 1). Quite surprisingly, DMT1 patients had significantly lower LDL and total cholesterol levels than controls. On the other hand, there were no differences in triglycerides or HDL-cholesterol

**Table 1.** Parameters comparison between the healthy and type 1 diabetes mellitus (DMT1) group.

| Variable | Healthy donors | DMT1 | p-value[2] |
|---|---|---|---|
| | (n = 50) | (n = 50) | |
| Male sex – number (%) | 26 (52) | 24 (48) | 0.689[3] |
| Age, years – mean (median, range) | 43 (44, 21-67) | 42 (41, 19-74) | 0.593 |
| BMI, kg/m$^2$ (mean ± SD) | 26.58 ± 3.86 | 27.11 ± 4.33 | 0.523[4] |
| Overweight or obesity – number (%) | 29 (58) | 34 (68) | 0.3[3] |
| Smoking – number (%) | 12 (24) | 11 (22) | 0.812[3] |
| COVID in the period of 2021–2022 – number (%) | 18 (36) | 27 (54) | 0.07[3] |
| Hypertension – number (%) | 9 (18) | 13 (26) | 0.334[3] |
| Hypothyroidism – number (%) | 6 (12) | 6 (12) | 1[3] |
| Glucose, mM (median, 95% CI) | **5.1 (4.9-5.3)** | **6.84 (6.17-7.8)** | **<0.001** |
| Glycated hemoglobin – mmol/mol (FGSP[1]) | not measured | 59 ± 12 (7.6 ± 3.3%) | – |
| LDL, mM (median, 95% CI) | **3.33 (3.09-3.52)** | **2.59 (2.37-2.92)** | **<0.001** |
| HDL, mM (median, 95% CI) | 1.45 (1.34-1.71) | 1.39 (1.24-1.54) | 0.200 |
| TG, mM (median, 95% CI) | 1.21 (1.04-1.36) | 1.00 (0.79-1.25) | 0.211 |
| Total cholesterol, mM (median, 95% CI) | **5.31 (5.05-5.69)** | **4.68 (4.27-5.07)** | **<0.001** |
| Non-HDL, mM (median, 95% CI) | **3.87 (3.61-4.01)** | **3.1 (2.72-3.43)** | **<0.001** |
| Vitamin K$_1$, nM (median, 95% CI) | **1.09 (0.87-1.48)** | **0.59 (0.38-1.04)** | **0.002** |
| Vitamin MK-4, nM (median, 95% CI) | **2.98 (2.68-3.23)** | **1.43 (0.97-1.94)** | **<0.001** |
| Vitamin MK-7, nM (median, 95% CI) | **0.39 (0.23-0.46)** | **0.49 (0.43-0.62)** | **0.008** |
| Total vitamin K, nM (median, 95% CI) | **4.47 (3.97-5.11)** | **2.70 (2.21-3.62)** | **<0.001** |
| Serum creatinine, mM (median, 95% CI) | 74.5 (69.3-81.8) | 75.8 (69.6-85.0) | 0.824 |
| Urine creatinine, mM (median, 95% CI) | 9.9 (8.4-13.4) | 10.0 (7.4-13.4) | 0.858 |
| Serum neopterin, nM (median, 95% CI) | **7.03 (6.51-8.07)** | **9.46 (8.18-10.98)** | **<0.001** |
| Urine neopterin/creatinine, nM/mM, (median, 95% CI) | **166 (153-194)** | **234 (200-255)** | **0.0004** |
| Serum kynurenine/tryptophan ratio, µM/mM (median, 95% CI) | **33.04 (31.5-37.08)** | **39.15 (36.86-44.88)** | **<0.001** |
| urine kynurenine/tryptophan ratio, µM/mM (median, 95% CI) | **55.27 (49.73-59.45)** | **71.94 (57.31-94.77)** | **0.004** |

Significant differences are marked in bold. CI, confidence interval

[1]The **N**ational **G**lycohemoglobin **S**tandardization **P**rogram units;

[2]sum of vitamin K$_1$ and vitamin K$_2$ (MK-4 and MK-7),

[2]The Mann-Whitney test was used if not specified otherwise,

[3]The Chi-squared test,

[4]Normality was confirmed in both healthy and DMT1 groups by the Shapiro-Wilk test, hence the T-test was used.

levels, serum and urine creatinine, BMI, or smoking habit. There were, however, clear differences in vitamin K levels: Levels of vitamin $K_1$ and vitamin $K_2$ form MK-4 were about twice higher in healthy subjects than in DMT1 patients, and on the contrary, the levels of another but much less abundant vitamin $K_2$ form MK-7 were in healthy subjects slightly but significantly lower than in DMT1 patients. Total vitamin K was also higher in controls than in DMT1 patients. All measured markers of inflammation (serum neopterin, urine neopterin to creatinine ratio, serum kynurenine to tryptophan ratio, and urine kynurenine to tryptophan ratio) were significantly higher in DMT1 patients.

Detailed characteristics of the patients and healthy individuals are specified in Table 1. The complete list of drugs used by healthy donors and DMT1 patients is attached in supplementary data (S1 Table).

## Coagulation profile of the enrolled subjects

Coagulation activity between healthy individuals and DMT1 patients was evaluated by means of PT (reported as INR) and aPTT (Fig 1). Basal values, e.g., from plasma solely treated with 1% of DMSO as the vehicle control, were not different between healthy controls and DMT1 patients in both assays. However, indirect anticoagulant heparin (the positive control) prolonged coagulation more intensively in DMT1 patients as compared to controls. This finding was observed in both the PT and aPTT tests. Similarly, dabigatran also prolonged coagulation more extensively in DMT1 patients than in generally healthy controls in both tests (Fig 1). In the case of xabans (factor Xa inhibitors), a higher potency was observed in DMT1 patient samples when PT values were compared. On the other hand, argatroban, the second direct thrombin inhibitor used in this experiment, had the same effect in both healthy and DMT1 patients. As there were differences in drug administration between groups (S1 Table), further analyses were performed by exclusion of ACEi inhibitors, statins, gabapentinoids (gabapentin or pregabalin), and $H_1$-antihistamines. The results of these analyses were, however, almost the same in all cases; the significance between generally healthy donors and DMT1 patients did not change (S2 Table).

## Effects of age and BMI on coagulation

In the following step, the influence of anthropological parameters, such as age and BMI, was investigated. Enrolled subjects were divided into groups of 10-year age ranges. When PT was analysed, heparin and dabigatran yielded significant differences between healthy and DMT1 patients in all age-groups, with the exception of solely a tendency (p = 0.054) in the 30–39 years group after heparin treatment (Fig 2A–2D). The effect of argatroban, the other thrombin inhibitor used,

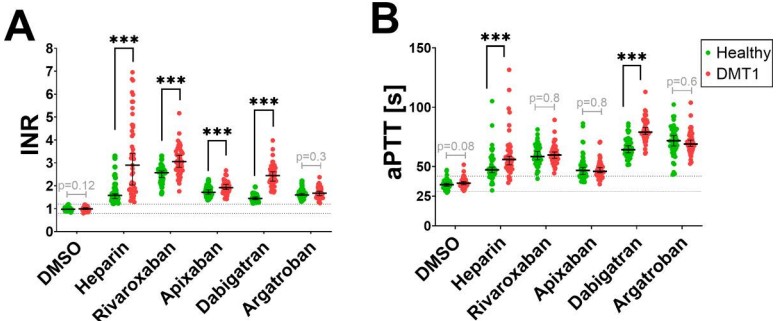

**Fig 1. Coagulation differences between healthy donors and DMT1 patients.** (A) prothrombin time (PT) expressed as international normalized ratio (INR), (B) activated partial thromboplastin time (aPTT). DMSO was used as the vehicle control with a final concentration of 1% and heparin (indirect anticoagulant as the positive control) at a final concentration of 5 and 0.5 IU/mL for PT and aPTT assays, respectively. The final concentration for all tested direct anticoagulants was 1 μM. Normality (based on the Shapiro-Wilk test) was observed for DMSO-INR, rivaroxaban-INR, apixaban-INR, and dabigatran-aPTT in both data sets, hence, the Student's t-test was employed. The Mann-Whitney test was carried out in all other cases. Data are shown as median (mean) with 95% confidence intervals. *** p < 0.001. Dotted lines represent physiological values of INR (0.8 to 1.2) and reference values of the manufacturer for aPTT (29 to 42 seconds).

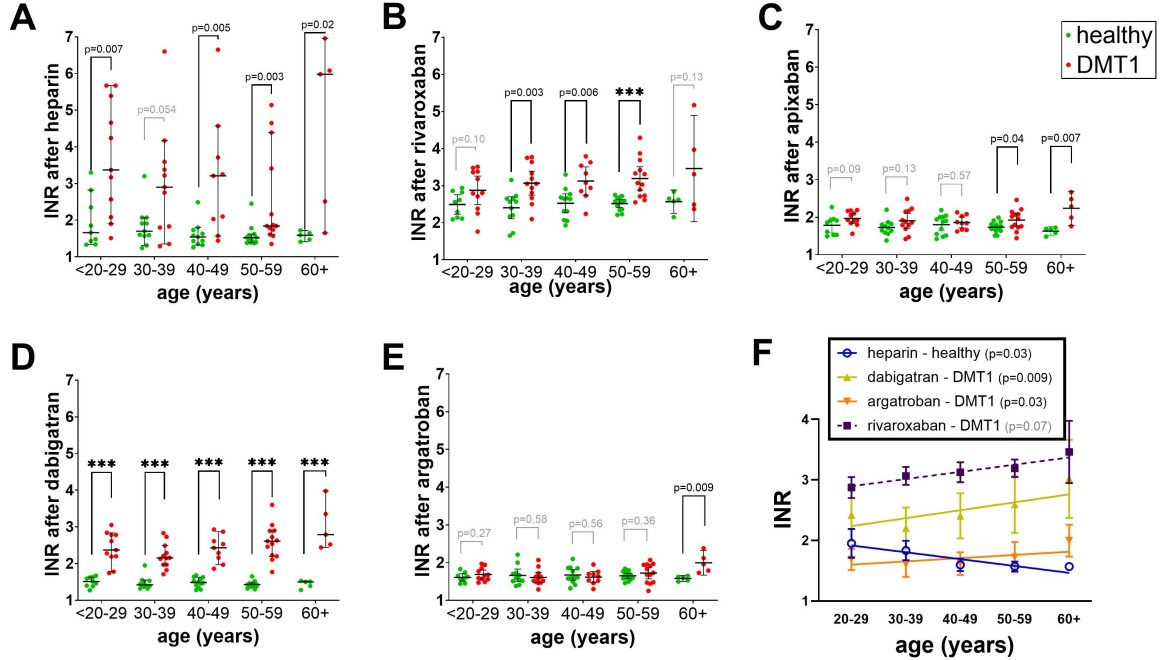

**Fig 2. Differences in prothrombin time (PT) based on the age ranges in healthy donors and DMT1 patients after treatment with anticoagulants.**
(A) heparin, (B) rivaroxaban, (C) apixaban, (D) dabigatran, (E) argatroban, and (F) linear regression between the effect and age in case of significant or nearly significant relationships. PT was expressed as international normalized ratio (INR). The Gaussian distribution was observed in most cases, and hence the unpaired Student's t-test was employed. In other cases (heparin age categories 30y+ and dabigatran 30-39y) the Mann-Whitney test was performed. Data are shown as mean (median) ± 95% confidence intervals. *** p < 0.001.

was much weaker than that of dabigatran, and yielded a significant difference only in the oldest patients enrolled. Regarding the activities of xabans, rivaroxaban yielded significant differences in coagulation profiles between healthy and DMT1 samples in age-ranges between 20 and 59, whereas apixaban caused different effects solely in the elder individuals (50–59 and >60; Fig 2B and 2C). Additionally, the effect of dabigatran, argatroban, and rivaroxaban followed an increasing trend with age in DMT1 patients but not in healthy volunteers (Fig 2F). Interestingly, the effect of heparin dropped linearly with age in healthy control donors but not in DMT1 patients (Fig 2F).

The effects observed in the aPTT test were different to those obtained with the PT (Fig 3). No differences between healthy volunteers and DMT1 patients were observed in most cases, with the exception of dabigatran (all age categories), and one age category (50–59 years) after heparin treatment. Although dabigatran was more potent in DMT1 patients in all age categories, a clear relationship between age and the effect was not observed, in contrast to the PT test. Heparin behaved similarly in relation to age in both the PT and aPTT tests, as its tendency to have a lower effect in healthy patients with increasing age was observed also in the aPTT assay (p = 0.07, Fig 3F). Unexpectedly, the effect of argatroban also dropped with age in healthy persons but not in DMT1 (Fig 3E and 3F).

A similar data analysis was performed with BMI (Fig 4 and S1 Fig). BMI was divided into 3 groups according to the international BMI index criteria: 18.5–24.9 (normal weight), 25–29.9 (overweight), and 30+ (obesity). Significant PT differences between healthy volunteers and DMT1 patient groups were observed in all BMI categories for all drugs tested with the exception of argatroban (Fig 4). In most cases, higher BMI was associated with lower anticoagulant effects, and this outcome was not limited to healthy patients, in contrast to age (Fig 4F). In case of aPTT (S1 Fig), the differences followed a similar pattern as in the age-effect analysis. They were, however, observed solely for dabigatran in all BMI categories and only in two BMI categories for heparin (18.5–24.9 and 30+).

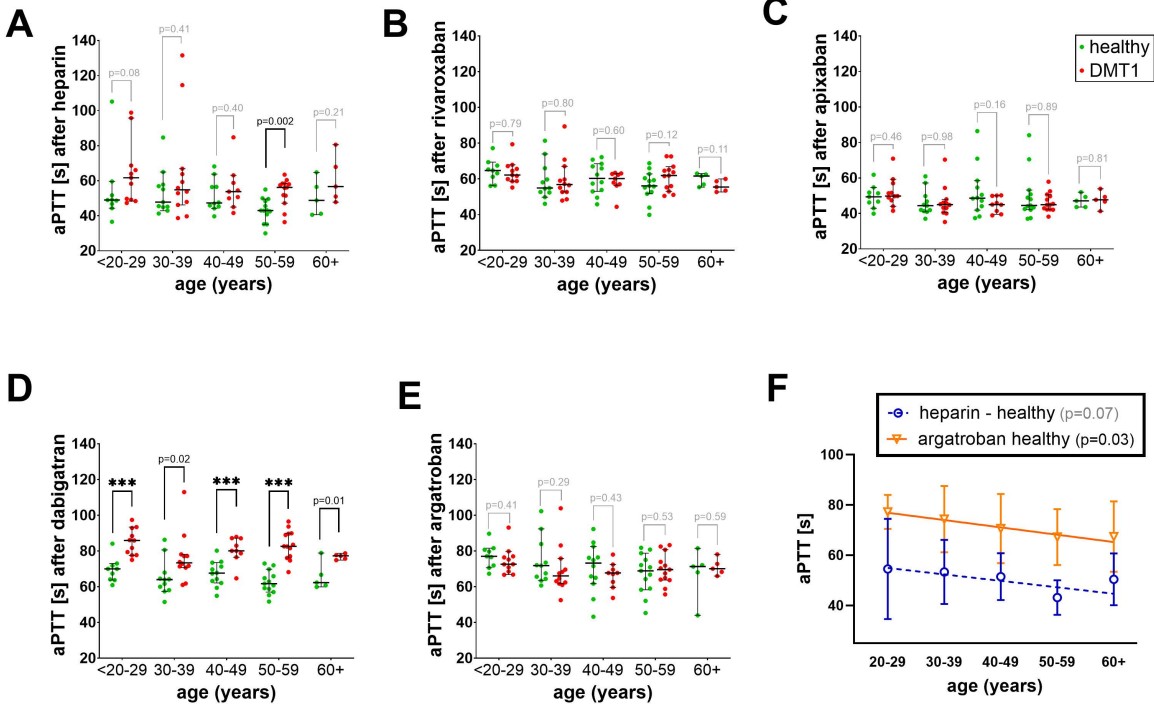

**Fig 3. Differences in activated partial thromboplastin time (aPTT) based on the age ranges in healthy donors and DMT1 patients after treatment with anticoagulants.** (A) heparin, (B) rivaroxaban, (C) apixaban, (D) dabigatran, (E) argatroban, and (F) linear regression between the effect and age in case of significant or nearly significant relationships. In all age groups 30-39y, heparin 20-29y, rivaroxaban 40-49y, and apixaban 50-59y groups, the Gaussian distribution was not confirmed, and hence the Mann-Whitney test was performed; in all other cases, the unpaired Student's t-test was employed. Data are shown as median ± 95% confidence intervals in all cases. *** p < 0.001.

Complementary analyses were performed in persons with diabetes, in which coagulation results were divided into two groups depending on glycemia, below or above 7 mM, and on glycated hemoglobin, below or above 53 mmol/mol. However, almost no differences were found. The sole exception was heparin, which was more active in DMT1 patients with glycated hemoglobin over 53 mmol/mol based on the PT test (S2 and S3 Figs).

## Correlations between biochemical parameters and coagulation parameters

In the next step, measured biochemical values were correlated with the obtained coagulation results (Table 2). In this analysis, logically all data (healthy controls and DMT1 patients) were included. PT values provided multiple significant correlations with several biochemical markers. LDL, non-HDL, and total cholesterol correlated both with PT and aPTT in most cases (S4 and S5 Figs). Correlation analysis provided negative values in these cases, meaning that higher levels, in particular of LDL or total cholesterol, shortened the coagulation time and hence decreased the effect of the tested anticoagulants. Interestingly, and similarly to the aforementioned, there was almost no correlation between glycemia and coagulation. Analogously, no correlations between glycated hemoglobin and coagulation in the whole group were observed. Glycated hemoglobin did not differ among age, gender, BMI or smoking status (S6 Fig), but some correlations were found in a subgroup analysis (S3 Table, S7 Fig). Total vitamin K correlated negatively mostly with PT. Therefore, higher levels of vitamin K led to shorter coagulation times. Subanalysis of vitamin K forms showed that the results were mostly driven by vitamin $K_1$ and MK-4 (S8 and S9 Figs). MK-7 form of vitamin $K_2$ did not correlate with any of the parameters detected. Contrarily, the same correlations were not observed with aPTT with two exceptions (solvent with total vitamin K and MK-4).

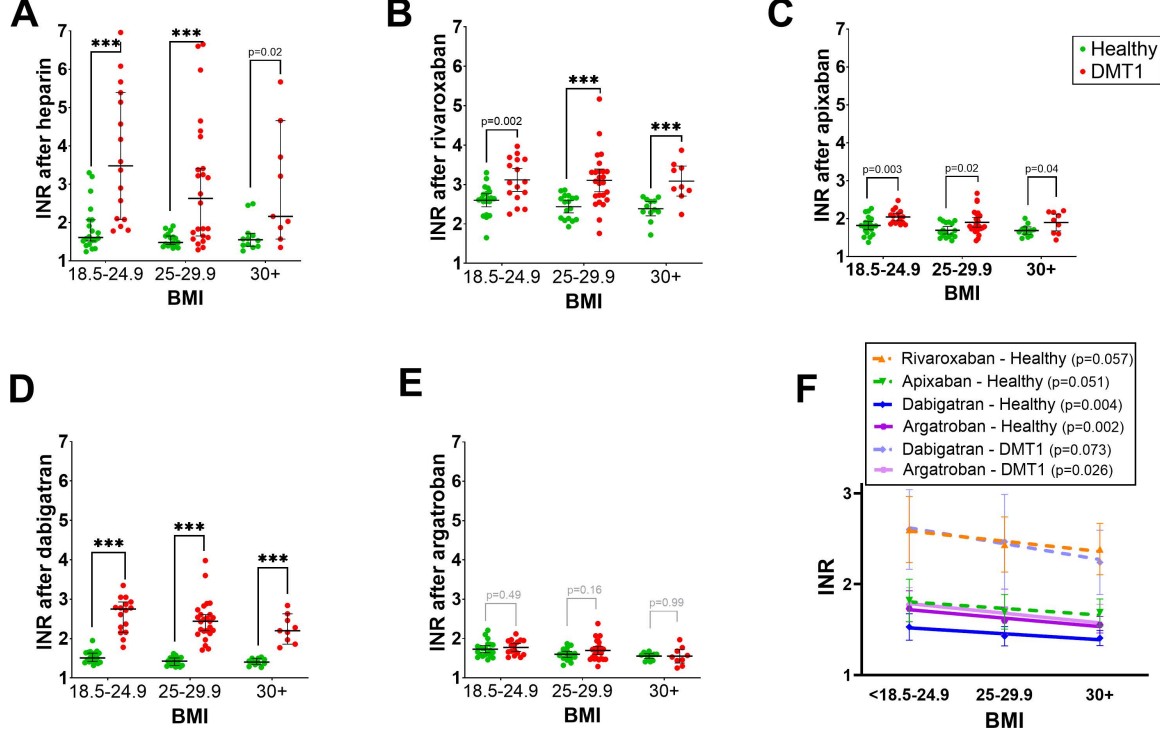

**Fig 4. Differences in prothrombin time (PT) based on the body mass index (BMI) range in healthy donors and DMT1 patients after treatment with anticoagulants.** (A) heparin, (B) rivaroxaban, (C) apixaban, (D) dabigatran, (E) argatroban, and (F) linear regression between the effect and BMI categories in case of significant or nearly significant relationships. PT was expressed as an international normalized ratio (INR). In case of all BMI heparin groups and overweight patients with dabigatran, a non-Gaussian distribution was found, hence, the Mann-Whitney test was employed. In other cases, the unpaired Student T-test was used. Data are shown as mean (median) ± 95% confidence intervals. *** $p < 0.001$.

## Correlation with inflammatory markers

Firstly, correlation analysis between inflammatory markers (serum neopterin; urinary neopterin/creatinine ratio, serum kynurenine/tryptophan and urinary kynurenine/tryptophan ratio) themselves, BMI, age, and biochemical markers was performed (Table 3). All four selected inflammatory markers correlated well together in a positive way. Two markers correlated also positively with age and BMI, while positive correlation with glycaemia and triglyceridemia was reported solely in one marker case. No correlation with LDL or non-HDL cholesterol was observed. HDL-cholesterol correlated negatively with serum neopterin, which means that higher neopterin equated to lower HDL-cholesterol. Interestingly, total vitamin K, vitamin $K_1$ or MK-4 correlated generally negatively with inflammatory markers, although the relationship did not always reach significance. Contrarily, MK-7 showed a positive relationship to serum neopterin.

Analysis with coagulation parameters showed significant correlations with inflammatory markers mainly for the effects of the most potent anticoagulant drugs, dabigatran and rivaroxaban (Fig 4). The anticoagulant effect of dabigatran correlated positively with all 4 inflammatory markers in the case of PT and with both systemic (serum) markers in the case of aPTT. Rivaroxaban showed a similar pattern with PT values (three correlations were significant and one nearly significant), but not with aPTT. Solvent (DMSO), heparin, and apixaban did not correlate at all with inflammatory markers, while one negative correlation was found for the effect of argatroban (aPTT, serum neopterin).

**Table 2. Correlations of selected biochemical profiles with the coagulation values.**

| | Parameters | DMSO | | Heparin | | Factor Xa inhibitors | | | | Thrombin Inhibitors | | | |
|---|---|---|---|---|---|---|---|---|---|---|---|---|---|
| | | | | | | Rivaroxaban | | Apixaban | | Dabigatran | | Argatroban | |
| | | r | p-value | r | p-value | r | p-value | r | p-value | r | p-value | r | p-value |
| INR | Glucose | −0.053 | 0.6 | 0.194 | 0.055 | 0.08 | 0.427 | 0.046 | 0.647 | **0.335** | **<0.001** | −0.068 | 0.503 |
| | gHb | −0.095 | 0.512 | 0.271 | 0.06 | 0.057 | 0.695 | 0.005 | 0.973 | **−0.143** | **0.323** | −0.122 | 0.398 |
| | LDL | **−0.227** | **0.023** | **−0.363** | **<0.001** | **−0.232** | **0.02** | **−0.452** | **<0.001** | **−0.515** | **<0.001** | **−0.335** | **<0.001** |
| | Total cholesterol | **−0.23** | **0.021** | **−0.313** | **0.002** | **−0.253** | **0.011** | **−0.465** | **<0.001** | **−0.479** | **<0.001** | **−0.321** | **0.001** |
| | HDL | 0.071 | 0.482 | −0.114 | 0.264 | −0.161 | 0.109 | 0.007 | 0.942 | −0.051 | 0.614 | 0.175 | 0.081 |
| | Triglycerides | **−0.237** | **0.017** | −0.036 | 0.723 | **−0.223** | **0.026** | **−0.308** | **0.002** | **−0.31** | **0.002** | **−0.441** | **<0.001** |
| | Non-HDL | **−0.246** | **0.014** | **−0.282** | **0.005** | **−0.215** | **0.032** | **−0.474** | **<0.001** | **−0.485** | **<0.001** | **−0.39** | **<0.001** |
| | Vitamin K$_1$ | **−0.257** | **0.011** | 0.046 | 0.657 | **−0.255** | **0.012** | −0.188 | 0.066 | **−0.347** | **<0.001** | −0.165 | 0.107 |
| | Vitamin MK-4 | **−0.298** | **0.003** | −0.149 | 0.145 | **−0.37** | **<0.001** | **−0.334** | **<0.001** | **−0.477** | **<0.001** | −0.135 | 0.183 |
| | Vitamin MK-7 | −0.108 | 0.288 | 0.023 | 0.822 | 0.046 | 0.652 | −0.115 | 0.255 | 0.17 | 0.093 | −0.117 | 0.249 |
| | Total vitamin K | **−0.357** | **<0.001** | −0.07 | 0.496 | **−0.384** | **<0.001** | **−0.37** | **<0.001** | **−0.458** | **<0.001** | **−0.225** | **0.024** |
| | Creatinine serum | −0.065 | 0.521 | −0.181 | 0.075 | 0.011 | 0.915 | −0.093 | 0.36 | −0.019 | 0.854 | −0.036 | 0.719 |
| | Creatinine urine | 0.035 | 0.732 | −0.096 | 0.346 | 0.047 | 0.644 | −0.034 | 0.739 | 0.036 | 0.723 | 0.092 | 0.362 |
| aPTT | Glucose | 0.163 | 0.105 | **0.252** | **0.012** | −0.106 | 0.296 | −0.189 | 0.06 | −0.098 | 0.33 | −0.06 | 0.556 |
| | gHb | −0.159 | 0.269 | 0.079 | 0.585 | −0.032 | 0.825 | −0.01 | 0.944 | 0.134 | 0.353 | −0.025 | 0.863 |
| | LDL | **−0.353** | **<0.001** | **−0.415** | **<0.001** | −0.142 | 0.16 | −0.074 | 0.466 | −0.137 | 0.175 | −0.179 | 0.074 |
| | Total cholesterol | **−0.403** | **<0.001** | **−0.399** | **<0.001** | −0.19 | 0.058 | −0.111 | 0.271 | −0.181 | 0.072 | **−0.199** | **0.048** |
| | HDL | 0.007 | 0.947 | 0.013 | 0.899 | −0.049 | 0.63 | 0.036 | 0.722 | −0.05 | 0.622 | −0.014 | 0.889 |
| | Triglycerides | **−0.317** | **0.001** | **−0.246** | **0.015** | −0.168 | 0.095 | **−0.256** | **0.01** | −0.128 | 0.206 | −0.123 | 0.222 |
| | Non-HDL | **−0.379** | **<0.001** | **−0.393** | **<0.001** | −0.148 | 0.143 | −0.138 | 0.171 | −0.159 | 0.113 | −0.187 | 0.063 |
| | Vitamin K$_1$ | −0.097 | 0.344 | −0.1 | 0.923 | 0.026 | 0.801 | −0.126 | 0.219 | −0.021 | 0.841 | 0.092 | 0.368 |
| | Vitamin MK-4 | **−0.269** | **0.007** | −0.188 | 0.065 | −0.173 | 0.087 | −0.193 | 0.055 | −0.144 | 0.154 | −0.165 | 0.103 |
| | Vitamin MK-7 | −0.144 | 0.154 | −0.039 | 0.707 | −0.049 | 0.63 | −0.114 | 0.263 | 0.097 | 0.341 | 0.091 | 0.368 |
| | Total vitamin K | **−0.276** | **0.006** | −0.149 | 0.144 | −0.09 | 0.371 | −0.185 | 0.066 | −0.108 | 0.284 | −0.081 | 0.426 |
| | Creatinine serum | 0.067 | 0.508 | −0.094 | 0.359 | −0.058 | 0.57 | −0.18 | 0.073 | −0.102 | 0.312 | −0.029 | 0.775 |
| | Creatinine urine | −0.025 | 0.802 | −0.06 | 0.558 | 0.155 | 0.124 | 0.048 | 0.633 | 0.144 | 0.153 | 0.092 | 0.361 |

Data were analyzed for both healthy and DMT1 patients together. gHb - glycated hemoglobin (HbA1c); LDL – low-density lipoprotein; HDL – high-density lipoprotein.

Correlation analyses were also performed based on the normality; in case of normal data distribution, the Pearson test was used; otherwise, the Spearman test was employed.

## The impact of glycemia, lactate, and hyperosmolarity on coagulation in acute settings

As untreated or poorly treated DMT1 is associated with hyperglycemia, lactate acidosis, and/or hyperosmolarity, complementary experiments in which the impact of these conditions on coagulation tests were carried out. In this series of tests, in addition to PT and aPTT, the thrombin test (TT) was also included.

In the first set of experiments, the acute impact of glucose mixed with plasma before the experiment was evaluated. In line with our population data, no impact of glucose up to the tested final concentrations of 30 mM on coagulation time was observed. When blood was incubated with glucose in a final concentration of 30 mM for 60 minutes, aPTT coagulation time was slightly but significantly prolonged in contrast to PT or TT. Lactate in different concentrations up to 10 mM had no impact on all three coagulation times. The addition of 20 mM of sodium chloride in order to mimic hyperosmolarity impacted aPTT and TT, although differently. Whereas aPTT was shortened, TT was prolonged (S10 and S11 Figs).

**Table 3. Correlations of selected inflammatory markers to other anthropological and biochemical parameters.**

| | Serum neopterin | serum kynurenine/ tryptophan | urinary kynurenine/ tryptophan | urinary neopterin-creatinine ratio | age | BMI | MK-4 | $K_1$ | MK-7 | total vitamin K | glucose | LDL | HDL-cholesterol | triglycerides | non-HDL cholesterol |
|---|---|---|---|---|---|---|---|---|---|---|---|---|---|---|---|
| serum neopterin | X | 0.521 (p<0.001) | 0.375 (p<0.001) | 0.630 (p<0.001) | 0.287 (p=0.004) | 0.267 (p=0.0075) | −0.142 (p=0.16) | 0.022 (p=0.83) | 0.29 (p=0.004) | −0.004 (p=0.97) | 0.207 (p=0.04) | 0.064 (p=0.53) | −0.319 (p=0.0013) | 0.212 (p=0.035) | 0.151 (p=0.14) |
| serum kynurenine/ tryptophan | 0.521 (p<0.001) | X | 0.529 (p<0.001) | 0.539 (p<0.001) | 0.191 (p=0.057) | 0.151 (p=0.13) | −0,2504 (p=0.011) | −0.303 (p=0.002) | 0.121 (p=0.22) | −0.263 (p=0.007) | 0.107 (p=0.29) | −0.078 (p=0.44) | −0.132 (p=0.19) | −0.059 (p=0.56) | −0.069 (p=0.50) |
| urinary kynurenine/ tryptophan | 0.375 (p<0.001) | 0.529 (p<0.001) | X | 0.323 (p=0.0014) | 0.310 (p=0.002) | 0.089 (p=0.38) | −0,258 (p=0.0091) | −0.165 (p=0.1) | 0,19 (p=0.057) | −0.150 (p=0.13) | −0,032 (p=0.75) | −0.077 (p=0.45) | −0.106 (p=0.3) | 0.04 (p=0.69) | −0.011 (p=0.91) |
| urinary neopterin-creatinine ratio | 0.630 (p<0.001) | 0.539 (p<0.001) | 0.323 (p=0.0014) | X | 0.202 (p=0.055) | 0.233 (p=0.027) | −0.131 (p=0.21) | −0.122 (p=0.25) | 0.097 (p=0.35) | −0.099 (p=0.34) | 0.113 (p=0.29) | 0.03 (p=0.78) | −0.148 (p=0.16) | 0.079 (p=0.46) | 0.023 (p=0.83) |

All inflammatory data sets did not demonstrate Gaussian distribution (tested by the Shapiro-Wilk test). Spearman correlation coefficients are hence reported in the table above.

## Discussion

Diabetes mellitus is a metabolic disease that can lead to a prothrombotic state and has a high incidence of major adverse cardiac events [7,8]. Moreover, patients suffering from this disease are at a higher risk of developing atrial fibrillation, which is a major risk for stroke [27]. Anticoagulation therapy, particularly DOACs, is now crucial in atrial fibrillation as they have better and safer pharmacokinetic and pharmacodynamic profiles than older anticoagulants [28]. In this study, we intentionally selected two direct thrombin inhibitors as well as two factor Xa inhibitors to enable comparison. These drugs are even more important in diabetes mellitus since hyperglycemia and insulin resistance increase thromboembolic risk through different processes such as hypercoagulability, endothelial dysfunction, or impaired fibrinolysis [4–6,29].

Insulin resistance and high glucose plasma levels exert synergistic effects on the extrinsic tissue factor or tissue factor pathway, leading to increased pro-coagulatory activity through an enhanced FVII conversion to its activated form (FVIIa) [30]. There are, indeed, studies reporting shortened PT (INR) and aPTT in patients suffering from type 2 diabetes mellitus. These differences are particularly pronounced in untreated or insufficiently treated patients with diabetes [31,32]. Our study has not found differences in coagulation times between DMT1 and control patients in untreated samples. Somehow unexpectedly, all INR values in both groups were in the physiological range 0.8–1.2. In the case of aPTT, which has physiological values based on the used test within the interval of 29–42 seconds, solely 4 DMT1 patients (8%) had longer aPTT while 5 controls (10%) were out of the range (2 shorter and 3 longer). Hence, it is not surprising that 1) no differences DMT1 *vs.* controls were observed and 2) relationships to other values were not observed or less frequently than in samples treated with anticoagulants. The likely explanation for no difference in both coagulation times is that our DMT1 patients were pharmacologically well managed not solely for diabetes but also for other risk factors, including dyslipidemia (Table 1, S1 Table). This result seems to be on one side paradoxical, but when we look at serum lipid levels in healthy and DMT1 donors employed in our study, this result is expected. DMT1 patients are logically more controlled and indeed, 11 of our DMT1 patients were treated with statins (S1 Fig) in contrast to none from the healthy control group, and they had lower LDL and total cholesterol levels (Table 1), plausibly for this reason. Although DMT1 type 2 diabetes mellitus patients have similar coagulation abnormalities, there are differences [6]. Also, the effect of hyperglycaemia and hyperinsulinemia on blood hemostasis is different [29]. Previous direct comparison of PT and aPTT values between healthy, DMT1, and type 2 diabetes mellitus patients confirmed our findings, as there were no differences between DMT1 and healthy controls while PT was prolonged in type 2 diabetes mellitus patients [33]. Interestingly, another recent study found a contrarily shortened aPTT in type 2 diabetes mellitus patients when compared to DMT1 patients, but no difference in PT [34].

Regardless, antidiabetic treatment seems to be able to normalize coagulation times [32] and indeed both aPTT and PT decrease (suggesting procoagulation) with increases in glycated hemoglobin or fasting plasma glucose [31]. In both mentioned studies, diabetes mellitus appeared to affect aPTT more extensively than PT values. This might suggest that the extrinsic coagulation pathway, which is reflected better by PT, is affected less extensively in patients suffering from type 2 diabetes mellitus. Our data with DMT1 patients showed, however, rather, a higher impact on PT, where the differences between healthy persons and DMT1 were more pronounced compared to those of aPTT (Fig 1A). This can be explained by a stronger effect of investigated anticoagulants on PT than aPTT: the most potent drug in the PT test rivaroxaban prolonged INR by 165% in the case of healthy controls and by 210% in diabetic patients *vs.* the solvent, while dabigatran, the most active compound in case of aPTT, increased aPTT less, by 82% and 120%, respectively. There were, however, differences in coagulation times among used anticoagulants. More potent drugs, rivaroxaban and dabigatran, had more associations than less potent apixaban and argatroban (Tables 2 and 4).

Although both anticoagulant tests have been used for decades, some aspects remain unclear. Specifically, the effect of thrombin inhibitors is better reflected by the aPTT test as by the PT assay, which is contrarily more sensitive to factor Xa inhibitors [20,35]. This can also explain the fact that factor Xa inhibitors, apixaban and rivaroxaban, prolonged significantly solely PT but not aPTT values when healthy donors and patients with diabetes were compared. Even if there are

Table 4. Correlations of selected inflammatory markers with results of coagulation tests.

| | DMSO INR | Heparin INR | Rivarox-aban INR | Apix-aban INR | Dabiga-tran INR | Argatro-ban INR | DMSO aPTT | Heparin aPTT | Rivar-oxaban aPTT | Apixaban aPTT | Dabiga-tran aPTT | Arga-troban aPTT |
|---|---|---|---|---|---|---|---|---|---|---|---|---|
| serum neopterin | 0.009 (p=0.93) | 0.046 (p=0.66) | **0.246 (p=0.014)** | −0.038 (p=0.71) | **0.263 (p=0.008)** | −0.038 (p=0.71) | −0.046 (p=0.65) | −0.013 (p=0.90) | −0.023 (p=0.82) | −0.139 (p=0.17) | **0.200 (p=0.048)** | **−0.203 (p=0.04)** |
| serum kynurenine/ tryptophan | 0.130 (p=0.20) | 0.154 (p=0.13) | **0.334 (p=0.0007)** | 0.106 (p=0.29) | **0.344 (p=0.0005)** | 0.070 (p=0.49) | −0.041 (p=0.69) | 0.047 (p=0.94) | 0.021 (p=0.84) | −0.031 (p=0.76) | **0.251 (p=0.012)** | −0.107 (p=0.29) |
| urinary kynurenine/ tryptophan | 0.084 (p=0.41) | 0.181 (p=0.078) | 0.187 (p=0.066) | 0.074 (p=0.47) | **0.345 (p=0.0005)** | 0.062 (p=0.55) | −0.078 (p=0.45) | 0.005 (p=0.96) | −0.111 (p=0.28) | −0.174 (p=0.087) | 0.160 (p=0.12) | −0.163 (p=0.11) |
| urinary neopterin-creatinine ratio | 0.133 (p=0.21) | 0.157 (p=0.14) | **0.207 (p=0.049)** | 0.023 (p=0.83) | **0.209 (p=0.047)** | 0.014 (p=0.89) | −0.143 (p=0.18) | 0.067 (p=0.53) | −0.173 (p=0.10) | −0.135 (p=0.201) | 0.092 (p=0.38) | −0.011 (p=0.92) |

All inflammatory data sets did not fit to Gaussian distribution pattern (tested by the Shapiro-Wilk test). Spearman correlation coefficients are hence reported in the table above.

mentioned differences in coagulation and pathophysiology between DMT1 and type 2 diabetes mellitus, it is not clear if the coagulation pattern is modified similarly in both types of patients. Regardless, hyperglycemia is common for both of them. Indeed, high blood glucose concentration impairs coagulation *via* both extrinsic and intrinsic pathways, both *in vivo* [36] and *in vitro* [37]. Additionally, it amplifies aggregation and adhesion of platelets through up-regulation of expression of pro-aggregatory factors like P-selectin, thromboxane $A_2$ ($TXA_2$), and von Willebrand factor (vWF) [5,38]. A systematic review and meta-analysis confirmed that patients suffering from DM have a higher risk of stroke or systemic embolism, cardiovascular death, but it also found that these patients have a higher risk of bleeding when treated with DOAC [39]. The latter outcome agrees well with our results (Fig 1).

The above-mentioned data suggest that higher glycemia can be the culprit. However, based on our correlation analysis, fasting glycemia itself was hardly the main factor able to explain the differences between healthy persons and DMT1 patients. There was almost no correlation between fasting glycemia and coagulation times in our study. Also, long-term glycemia measured indirectly by glycated hemoglobin (Table 1, S2 Table) was not able to explain the coagulation differences. Contrarily, differences in cholesterol seemed to be much more important. Higher cholesterol values are related to accelerated blood coagulation [40], an effect which was also observed in our study. Coagulation factors bind to lipophilic membranes and membrane fluidity, which is well known to be impacted by cholesterol, is one of the important factors for the coagulation process [41]. Similarly, coagulation factor VII activity is dependent on the intake of lipids [42]. Hence, the most likely explanation of our results can reside in better lipidic profiles of our DMT1 persons due to intense medical care over healthy controls.

As vitamin K is an important player in the synthesis of 7 coagulation and anticoagulation factors [15], its relationship with coagulation times was expected too. Higher levels of major vitamin $K_2$ form MK-4 were associated with shortening of coagulation times in general. This was observed in solvent-treated blood samples in both coagulation tests as well as for three of four direct anticoagulants in the PT test (S8 and S9 Figs). The relationship between vitamin K and diabetes is not sufficiently explored, but the biological effect of vitamin K might be beyond coagulation [15,43,44]. For example, vitamin K administration results in a decrease in fasting glucose based on a very recent meta-analysis of 7 studies. The effect was observed in type 2 diabetes mellitus patients, but not in non-diabetic persons, and vitamin K supplementation was also associated with a lower risk of developing type 2 diabetes mellitus [45]. This can suggest that persons with diabetes need more vitamin K which appears to be in accordance with our results showing lower levels of vitamin K in DMT1 patients.

Indeed, a recent analysis of vitamin $K_1$ levels suggested that its serum levels were mildly associated with micro- or macroangiopathy [46]. Authors did not report precise serum levels of vitamin $K_1$ in healthy persons and patients with diabetes, but reported the data according to the presence or absence of these complications with median levels ranging from 0.86 to 0.95 nM. The advantage of that study is that it included a much larger portion of patients in comparison to our population, but they did not measure vitamin $K_2$ forms and included also patients being treated with vitamin K anticoagulants, which can bias the results. Our study hence brought novel data in relation to vitamin K forms in coagulation. In addition to their well-reported effects on coagulation, vitamin K has anti-inflammatory effects, and recent data in accordance with the aforementioned suggested that it might modify glycaemia. The mechanism can be linked with the newly discovered pancreatic vitamin K-dependent protein called endoplasmic reticulum Gla protein [15,43,44]. The observed negative relationships between vitamin K levels and inflammatory markers (Table 3) seem to confirm the anti-inflammatory properties of this vitamin. Lower vitamin K levels in diabetic persons might suggest a higher need for vitamin K for normalization of glycaemia in DMT1 (Table 1). Another very novel finding is the observation of higher levels of MK-7, a minor form of vitamin $K_2$, in DMT1 patients and at the same time the absence of negative correlation with inflammatory markers.

Other factors which we evaluated in more detail were age and BMI values. It is commonly recognized that the physiological processes of aging coincide with a rise in the plasma concentrations of certain coagulation factors, including fibrinogen, factor V, factor VII, factor VIII, factor IX, high molecular-weight kininogen, and prekallikrein [47]. Such findings match with a decreasing effect of heparin in relation to increasing age in healthy controls (Figs 2 and 3) but do not correspond to a higher effect of some direct anticoagulants observed in DMT1 patients in the PT assay (Fig 2). Interestingly, high BMI values appear to be associated with lower effects of tested anticoagulants (Fig 4). This is quite an interesting finding as this was not an *in vivo* but *ex vivo* study, where the concentration was the same in all blood samples. This points to BMI-associated differences in the coagulation cascade not related to different volumes of distribution. As higher BMI is known to be associated with proinflammatory states [48], and this was observed in our study, too (Table 3), BMI was expected to predict rather longer coagulation times. In contrast to BMI, inflammatory markers were positively correlated with the prolongation of coagulation mediated by the most active anticoagulants, dabigatran and rivaroxaban (Table 4), suggesting that the relationship between BMI, inflammation, and coagulation is more complex. As the effect of anticoagulants in this study was stronger in diabetic persons while BMI tended to decrease the potency of the tested anticoagulant, it appears that BMI, in contrast to inflammation, does not seem to be one of the crucial factors responsible for the observed differences between healthy controls and DMT1. Contrarily, the impact of age cannot be fully rejected due to the fact that inflammation correlated with age, increased the effect of tested anticoagulants (Table 4), and age did or tended to have the same relationship at least in the case of PT (Fig 4). On the other hand, both groups were well age-balanced and hence the impact of age on the observed difference was likely not a key factor. The differences might hence rather correspond to an inflammatory state.

Inflammation, even of low degree is an independent risk factor in cardiovascular diseases. In this study, we selected relatively novel markers with cardiovascular importance. Neopterin was shown to be an independent predictor of all-cause and cardiovascular mortality regardless of the presence or absence of coronary artery disease [49]. Similarly high kynurenine/tryptophan ratio was documented in cardiovascular diseases [50,51]. Although one can expect that inflammation will rather cause procoagulation, our data suggested that higher inflammation is associated with improved effect of direct anticoagulants (Table 4). To support this unexpected finding, it should be emphasized that in severe COVID-19 patients, coagulation times (PT, aPPT) but also the level of fibrinogen correlated positively with another inflammatory marker, C-reactive protein, and this logically implicates that inflammation was related to prolonged coagulation [11].

Another aspect or even confounding factor can be the kidney function, which can have an impact on coagulation [52] and can be impaired in DMT1. However, no differences between serum and urine creatinine were detected in our study, suggesting no major differences in kidney function. Hence, this factor hardly played a role in the data obtained in our study and can be excluded.

We should recognize some limitations of our study: 1) We have included a relatively small sample size of patients and healthy donors. This study was intended as the first screening study to see if there can be at least a 10% difference in coagulation parameters in the whole tested set of DMT1 patients and generally healthy donors. Subgroup analyses included a lower number of samples, and confirmation of such data will need further studies. 2) Only one-time point measurement was carried out. Coagulation might be in principle affected by other factors which can be changed in time, e.g., de/hydration, etc., and paired samples could limit such short-term influences. 3) We have not assessed dietary habits as they might be different due to lifestyle recommendations for diabetic patients. 4) Due to limits in the volume of blood sampling, we were not able to include other relevant parameters (e.g., fibrinogen, D-dimer).

In summary, the observed differences in coagulation between generally healthy controls and DMT1 patients in our study were mediated most probably by differences in blood cholesterol, inflammatory state, and vitamin K levels. Glycemia, both short-term (fasting glycemia and *ex vivo* experiments where plasma was incubated with glucose) or long-term (assessed by glycated hemoglobin), by itself did not have a clear or decisive effect on coagulation times. Kidney function, age, gender, and BMI were well balanced between both groups and they were not the key factors explaining the differences. The differences in vitamin K levels require additional investigation as they might be a secondary process based on the DMT1 progression.

## Conclusions

To the best of our knowledge, our study is the first one that reported 1) a direct comparison of thrombin and factor Xa inhibitors in patients with DMT1, and 2) plasma levels of three major forms of vitamin K in this group in relation to coagulation. Based on our results, the tested anticoagulants seem to be more effective in DMT1 patients than in healthy individuals. Three major reasons were identified: differences in blood cholesterol, vitamin K levels, and degree of inflammation. Both former two parameters were lower in DMT1 patients, while all 4 inflammatory markers measured were lower in healthy controls. Contrarily, we suppose that age, BMI, gender, glycemia, concomitantly administered drugs, and kidney function were not substantially involved in these differences. Our findings suggest that direct anticoagulants, in particular dabigatran and rivaroxaban, should be administered more carefully in patients suffering from DMT1 and possible dose adjustment should be performed.

## Supporting information

**S1 Fig. Differences in activated partial thromboplastin time (aPTT) based on the BMI ranges in healthy donors and DMT1 patients after treatment with anticoagulants.**
(TIF)

**S2 Fig. Comparison of (A) prothrombin time (PT, expressed as INR: international normalized ratio) and (B) activated partial thromboplastin time (aPTT) values for the negative control (solvent DMSO) and all tested anticoagulants based on the fasting levels of glucose.**
(TIF)

**S3 Fig. Comparison of (A) prothrombin time (PT, expressed as INR: international normalized ratio) and (B) activated partial thromboplastin time (aPTT) values for the negative control (solvent DMSO) and tested anticoagulants in relation to the levels of glycated haemoglobin in recruited type 1 diabetes mellitus patients.**
(TIF)

**S4 Fig. Correlation among prothrombin time (PT, expressed as INR: international normalized ratio) and selected biochemical parameters.**
(TIF)

**S5 Fig. Correlation between activated partial thromboplastin time (aPTT) and selected biochemical parameters.**
(TIF)

**S6 Fig. Comparison of glycated hemoglobin in relation to age (A), sex (B), BMI (C), and smoking habit (D) in type 1 diabetes mellitus patients.**
(TIF)

**S7 Fig. Correlations of the effect of tested anticoagulants (reported as INR or aPTT) with HbA1C in type 1 diabetes mellitus patients.**
(TIF)

**S8 Fig. Correlation between prothrombin time (PT, expressed as INR: international normalized ratio) and vitamin $K_1$ (A) and (B) vitamin $K_2$ form MK-4.**
(TIF)

**S9 Fig. Correlation between activated partial thromboplastin time (aPTT) and vitamin $K_2$ form MK-4.**
(TIF)

**S10 Fig. Impact of short-term (3 minutes, A-C) and long-term (60 minutes, D-F) treatment with glucose.**
(TIF)

**S11 Fig. Impact of lactate (A-C) and hyperosmolarity (20 mM sodium chloride, D-F) treatment.**
(TIF)

**S1 Table. Drugs used in generally healthy controls and patients suffering from type 1 diabetes mellitus.**
(DOCX)

**S2 Table. Statistical analysis of differences in coagulation parameters after exclusion of drug groups which were more prevalently administered either in the case of type I diabetes mellitus patients or healthy controls.**
(DOCX)

**S3 Table. Correlation of glycated haemoglobin from DMT1 patients – subgroup analysis.**
(DOCX)

## Author contributions

**Conceptualization:** Alejandro Carazo, Přemysl Mladěnka.

**Data curation:** Alejandro Carazo, Jaka Fadraersada, Přemysl Mladěnka.

**Formal analysis:** Alejandro Carazo, Jaka Fadraersada, Přemysl Mladěnka.

**Funding acquisition:** Alejandro Carazo, Pavel Skořepa, Lenka Kujovská Krčmová, Přemysl Mladěnka.

**Investigation:** Alejandro Carazo, Markéta Paclíková, Jaka Fadraersada, Raúl Alva-Gallegos, Pavel Skořepa, Catherine Gunaseelan, Kateřina Matoušová, Kristýna Mrštná, Lenka Kujovská Krčmová, Alena Šmahelová, Přemysl Mladěnka.

**Methodology:** Alejandro Carazo, Vladimír Blaha, Přemysl Mladěnka.

**Supervision:** Alejandro Carazo, Lenka Kujovská Krčmová, Vladimír Blaha, Přemysl Mladěnka.

**Writing – original draft:** Alejandro Carazo, Jaka Fadraersada, Přemysl Mladěnka.

**Writing – review & editing:** Alejandro Carazo, Markéta Paclíková, Jaka Fadraersada, Raúl Alva-Gallegos, Pavel Skořepa, Catherine Gunaseelan, Kateřina Matoušová, Kristýna Mrštná, Lenka Kujovská Krčmová, Alena Šmahelová, Vladimír Blaha, Přemysl Mladěnka.

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
