## [Decision Letter · Decision Letter 0]

PONE-D-24-43881Type 1 diabetes mellitus patients had lower total vitamin K levels and increased sensitivity to direct anticoagulantsPLOS ONE

Dear Dr. Mladěnka,

Thank you for submitting your manuscript to PLOS ONE. After careful consideration, we feel that it has merit but does not fully meet PLOS ONE’s publication criteria as it currently stands. Therefore, we invite you to submit a revised version of the manuscript that addresses the points raised during the review process.

We look forward to receiving your revised manuscript.

Kind regards,

Yousef Khazaei Monfared

Academic Editor

PLOS ONE

Journal Requirements:

This work was supported by the Czech Health Research Council (NU21J-02-00021); the project New Technologies for Translational Research in Pharmaceutical Sciences /NETPHARM, project ID CZ.02.01.01/00/22_008/0004607, co-funded by the European Union/, Charles University (SVV 260 549) and MH CZ - DRO (grant number UHHK, 00179906).

This work was supported by the Czech Health Research Council (NU21J-02-00021); the project New Technologies for Translational Research in Pharmaceutical Sciences /NETPHARM, project ID CZ.02.01.01/00/22_008/0004607, co-funded by the European Union/, Charles University (SVV 260 549) and MH CZ - DRO (grant number UHHK, 00179906).

This work was supported by the Czech Health Research Council (NU21J-02-00021); the project New Technologies for Translational Research in Pharmaceutical Sciences /NETPHARM, project ID CZ.02.01.01/00/22_008/0004607, co-funded by the European Union/, Charles University (SVV 260 549) and MH CZ - DRO (grant number UHHK, 00179906).

5. We note that you have indicated that there are restrictions to data sharing for this study. For studies involving human research participant data or other sensitive data, we encourage authors to share de-identified or anonymized data. However, when data cannot be publicly shared for ethical reasons, we allow authors to make their data sets available upon request. For information on unacceptable data access restrictions, please see http://journals.plos.org/plosone/s/data-availability#loc-unacceptable-data-access-restrictions.

6. In the online submission form, you indicated that some data cannot be shared publicly because of personal information. Data are stored on serves of the Faculty of Pharmacy, Charles University and some non-confidential data can be obtained upon a resonable request to the corresponding author.

7.  Please amend your list of authors on the manuscript to ensure that each author is linked to an affiliation. Authors’ affiliations should reflect the institution where the work was done (if authors moved subsequently, you can also list the new affiliation stating “current affiliation:….” as necessary).

Reviewers' comments:

Reviewer's Responses to Questions

**Comments to the Author**

1. Is the manuscript technically sound, and do the data support the conclusions?

Reviewer #1: Yes

Reviewer #2: Partly

Reviewer #3: No

Reviewer #4: Yes

Reviewer #5: Partly

2. Has the statistical analysis been performed appropriately and rigorously? 

Reviewer #1: I Don't Know

Reviewer #2: No

Reviewer #3: Yes

Reviewer #4: Yes

Reviewer #5: Yes

3. Have the authors made all data underlying the findings in their manuscript fully available?

Reviewer #1: Yes

Reviewer #2: Yes

Reviewer #3: No

Reviewer #4: No

Reviewer #5: Yes

4. Is the manuscript presented in an intelligible fashion and written in standard English?

Reviewer #1: Yes

Reviewer #2: No

Reviewer #3: Yes

Reviewer #4: Yes

Reviewer #5: Yes

5. Review Comments to the Author

Reviewer #1: Goodpaperwith enough information. However there is an important issue. For these outcome measures, the sample size is so limited. Because of lots of dropouts , it was needed to have more participants. You did not mention how to estimate sample size in method section. Moreover, I did not see the limitation part.

Reviewer #2: This manuscript, titled "Type 1 Diabetes Mellitus Patients Had Lower Total Vitamin K Levels and Increased Sensitivity to Direct Anticoagulants," explores an intriguing and clinically significant intersection between diabetes mellitus, coagulation pathways, and the efficacy of anticoagulants. The study aims to shed light on potential differences in anticoagulant sensitivity between patients with type 1 diabetes mellitus (DMT1) and healthy individuals, while also examining the role of vitamin K levels in mediating these differences.

The research is notable for its direct comparison of different classes of anticoagulants and its attempt to correlate biochemical markers with coagulation outcomes. However, the manuscript, as submitted, has several areas that warrant substantial revision. Addressing these issues will not only enhance the scientific rigor of the work but also its potential impact on clinical practice and public health.

The following detailed review outlines specific recommendations for improving the manuscript:

Abstract

p. 2, lines 25–46:

The abstract lacks a concise statement of the clinical or public health relevance of the findings. How these findings might influence current clinical practices is not well articulated.

Terms like "paradoxically better lipid profiles" need clarification; such phrasing is ambiguous without context.

The results are described without confidence intervals, p-values, or other statistical measures, which are essential to substantiate claims.

Introduction

p. 3, lines 61–66:

The introduction does not sufficiently highlight gaps in existing literature to justify the study’s necessity. It assumes the audience is already aware of the limitations in current research on DMT1 and anticoagulation.

p. 3, lines 66–70:

Statements regarding the suitability of direct anticoagulants for DMT1 patients are broad and unsupported by specific prior studies or references.

p. 3, lines 78–90:

The description of coagulation assays lacks precision. Specific references to how these tests uniquely contribute to the study goals could strengthen the argument.

Methods

p. 5, lines 99–102:

No rationale is provided for choosing 50 participants per group. Was a power calculation conducted to ensure adequate sample size?

p. 5, lines 101–105:

The criteria are described but lack justification. For instance, why were individuals with hypothyroidism or hypertension included if these conditions could influence coagulation?

Control Selection:

The control group, described as “generally healthy,” is not defined rigorously. Are they comparable to DMT1 patients in lifestyle and diet?

Confounding Variables:

The study fails to address the influence of potential confounders such as inflammatory markers, diet, or medication use.

Results

p. 8–12:

The narrative description of results omits some key statistical details. For example, confidence intervals for effect sizes are not reported.

p. 9–12:

There is minimal discussion of non-significant findings. This could introduce bias in interpreting the data.

Ambiguity in Causality:

Correlations between vitamin K levels and coagulation outcomes are mentioned, but causation is implied without robust evidence.

Demographic Stratification:

While stratified analyses by age and BMI are conducted, their clinical implications are not clearly discussed.

Discussion

p. 14–17:

Claims regarding the protective effects of lower cholesterol and vitamin K levels in DMT1 lack robust evidence. The discussion overstates the findings’ implications.

Confounding Factors:

The potential influence of unmeasured variables (e.g., inflammatory states or micronutrient deficiencies) is not adequately acknowledged.

Comparison to Prior Studies (p. 14):

The discussion does not sufficiently critique or compare the study's results with contradictory findings from other research.

Limitations (p. 17):

Limitations are inadequately addressed. For example, no mention of the small sample size or lack of long-term follow-up is made.

p. 18:

The conclusion is overly broad and lacks actionable insights for clinical practice.

Reviewer #3: The authors performed an interesting study. There are potential issues in methodology and thus interpretation of results. Specifically, I have the following comments:

1. I am not sure whether PT/APTT are regularly used to assess the prothrombotic burden

2. Similarly, PT/APTT may not be the standardised way to measure the anticoagulation effect of DOACs

3. Similarly, APTT instead of PT should be used to measure the effect of heparin

4. As the authors have noted, it was quite surprising that DMT1 individuals had lower levels of total cholesterol and LDL. Any possible explanation?

5. It was unclear why the authors performed subgroup analysis according to BMI. More elaboration is needed

6. Did the authors perform analysis correlating the strength of (anti-)coagulation and markers of insulin resistance?

7. Were there studies reporting an association between shortened PT/APTT and atherosclerotic risk?

8. Testing of anticoagulant level seems clinically more relevant than testing the effect at equimolar concentrations

9. The authors may consider performing a mediation analysis to support the findings

10. Please elaborate more on potential clinical implications and limitations

Reviewer #4: 1. Grammatical and Orthographic Review:

The manuscript is largely well-written with minor grammatical issues and awkward phrasing in some places. These areas could benefit from small revisions for clarity and flow:

Abstract: The sentence "There is, however, no effect comparison among different direct anticoagulants" could be more clearly phrased as "However, no comparative studies have been conducted on the effects of different direct anticoagulants."

Introduction: The transition from general coagulation background to diabetes is slightly abrupt. A smoother transition, linking diabetes-specific coagulation issues with broader coagulation principles, would improve clarity.

Methods: The description of chemical treatments and drug concentrations is concise, but the sentence "Heparin and dabigatran treatment resulted in longer coagulation in DMT1 when compared to healthy individual in both tests" can be improved to "Heparin and dabigatran treatments resulted in prolonged coagulation in DMT1 patients compared to healthy individuals in both tests."

Figures and Tables: Ensure that figure legends are consistently formatted. For example, "heparin (final concentration of 5 IU/mL)" could be "Heparin (5 IU/mL final concentration)" for consistency.

Results: In the sentence "There were however clear differences in vitamin K levels," adding a comma after "were" would enhance readability: "There were, however, clear differences in vitamin K levels."

2. Scientific Rigor:

Design and Methodology: The study is well-designed, using both control and experimental groups, and applies standard biochemical methods. However, it could benefit from more detailed information regarding the statistical tests used for each analysis. For example, when reporting results from the Student's t-test or the Mann-Whitney test, it's important to note the specific assumptions or conditions under which each test was applied, particularly since they differ in handling data distribution.

Cohort: The study is limited to a sample size of 50 DMT1 patients and 50 healthy controls. While this is a reasonable sample size for pilot studies, larger cohorts may be required to validate the findings robustly and examine variability across different subgroups (age, gender, comorbidities).

Data Presentation: The data is presented well in tables and figures, but ensuring that all statistical analyses are well-documented in the methods section (for reproducibility) is crucial. The authors might also consider adding confidence intervals for their estimates, which would provide more transparency and insight into the statistical reliability of their findings.

Vitamin K Levels: The study finds that DMT1 patients have lower total vitamin K levels compared to healthy controls, which is an intriguing result. However, the authors could further discuss the possible mechanisms behind this, referencing other literature on vitamin K and diabetes, especially in terms of bioavailability or altered metabolic pathways in DMT1 patients.

3. Scientific Impact:

The paper provides valuable insights into how DMT1 patients may experience altered coagulation and increased sensitivity to anticoagulants. The finding that vitamin K may play a role in coagulation disorders within DMT1 patients has significant potential for both clinical applications and further research.

The study also opens the door for exploring the relationship between lipid profiles, vitamin K, and anticoagulation therapies in DMT1 patients. This could have important implications for managing cardiovascular risks in these patients.

4. Questions for Authors and Further Research:

On Vitamin K and DMT1: What mechanisms do you propose for the observed lower levels of vitamin K in DMT1 patients? Could this be linked to altered dietary intake, absorption, or metabolic processing of vitamin K in these individuals?

On Lipid Profiles: The study found a paradoxical better lipid profile in DMT1 patients. Can you discuss in more detail how lipid metabolism in DMT1 may influence the coagulation system? Are these findings consistent with other studies, and how might they impact the broader therapeutic approach for managing DMT1?

On Anticoagulant Use: While the study shows increased sensitivity to direct anticoagulants in DMT1 patients, what are the potential clinical risks or benefits of using these anticoagulants in this population? Do the authors suggest any modifications to current anticoagulant dosing practices for DMT1 patients, considering their altered coagulation profiles?

On Age and BMI Effects: How do the effects of age and BMI on coagulation profiles compare to other factors such as lipid profiles or medication adherence in DMT1 patients? Could these factors confound the results, and how might they be better controlled in future studies?

Literature Gaps: There seems to be a lack of studies directly comparing coagulation in DMT1 patients with those of other diabetic subtypes (e.g., type 2 diabetes). Could the authors expand on the differences or similarities in coagulation patterns between DMT1 and type 2 diabetes, citing other relevant research?

5. Final Recommendation:

The manuscript presents an intriguing and relevant exploration of the coagulation profiles in type 1 diabetes mellitus patients, particularly focusing on the role of vitamin K and its impact on anticoagulation sensitivity. However, I recommend addressing the minor grammatical issues and providing additional clarity on the mechanisms behind the findings. Expanding the discussion on the clinical implications of these results, particularly for anticoagulant treatment in DMT1 patients, would enhance the manuscript's impact.

Reviewer #5: The study by Carazo A et al, titled “Type 1 diabetes mellitus patients had lower total vitamin K levels 2 and increased sensitivity to direct anticoagulants” aims to evaluate the importance vitamin K/ lipids in Type 1 diabetic coagulopathy. They have shown that anticoagulant treatment increases the coagulation time in patients with diabetes mellitus. While I find this finding intriguing, I have some concerns about the article.

Table 1: I wondered why the author did not measure glycated hemoglobin in healthy donors.

Author treated plasma with glucose and measured aPTT, PT, and TT. It would be interesting to treat plasma with triglycerides and cholesterol, and then measure the same parameters (aPTT, PT, and TT).

What is the D-dimer concentrations in healthy donors compared to patients with diabetes mellitus? It would be worthwhile to correlate glucose, glycated hemoglobin, LDL, HDL, and TAG levels with D-dimer concentrations.

Table 2: I am interested in understanding how the external addition of heparin or Factor Xa inhibitors affects the r/p-value compared to DMSO alone (with glucose).

Authors should specify the type of statistical analysis used in the figure legend.

6. PLOS authors have the option to publish the peer review history of their article (what does this mean? ). If published, this will include your full peer review and any attached files.

**Do you want your identity to be public for this peer review?** For information about this choice, including consent withdrawal, please see our Privacy Policy .

Reviewer #1: **Yes: ** Laleh Abadi marand

Reviewer #2: No

Reviewer #3: No

Reviewer #4: No

Reviewer #5: No

---

## [Author Response · Author response to Decision Letter 1]

11 Apr 2025

Reviewer #1: Good paper with enough information. However there is an important issue. For these outcome measures, the sample size is so limited. Because of lots of dropouts, it was needed to have more participants. You did not mention how to estimate sample size in method section. Moreover, I did not see the limitation part.

Response: We understand and agree with the points raised by the reviewer that the number of patients is a limitation of our paper. We are aware that a population of 50 patients and 50 healthy controls is relatively small, but we have estimated sample size: The number of patients (50 in each group) should be sufficient to prove a difference of 10% in anticoagulant response (power analysis: p=0.05, power 0.95 → 26 patients in each group are needed). In fact, we have recruited 50 patients due to potential dropouts in some subanalyses. These results are a part of a screening project where we have included patients with different metabolic disorders to see if there are differences. Recruitment of more patients and healthy donors would require novel ethics committee approval (as the last one ended with the end of the project in December 2024) and probably will not bring too much novel findings (mostly there are clear differences in the coagulation effects or there are apparently no differences, e.g. Figure 1). We suppose that, given the observed differences, the number of recruited blood donors was sufficient for this publication. Regardless, we newly specified power analysis in our paper (in the chapter on statistical analysis), and we have also added a new chapter on limitations.

Reviewer #2: This manuscript, titled "Type 1 Diabetes Mellitus Patients Had Lower Total Vitamin K Levels and Increased Sensitivity to Direct Anticoagulants," explores an intriguing and clinically significant intersection between diabetes mellitus, coagulation pathways, and the efficacy of anticoagulants. The study aims to shed light on potential differences in anticoagulant sensitivity between patients with type 1 diabetes mellitus (DMT1) and healthy individuals, while also examining the role of vitamin K levels in mediating these differences. The research is notable for its direct comparison of different classes of anticoagulants and its attempt to correlate biochemical markers with coagulation outcomes. However, the manuscript, as submitted, has several areas that warrant substantial revision. Addressing these issues will not only enhance the scientific rigor of the work but also its potential impact on clinical practice and public health.

Response: We are thankful for a very detailed assessment of our paper as well as for encouraging comments. Below are detailed answers to all comments.

The following detailed review outlines specific recommendations for improving the manuscript:

Abstract

p.2, lines25–46:

The abstract lacks a concise statement of the clinical or public health relevance of the findings. How these findings might influence current clinical practices is not well articulated. Terms like "paradoxically better lipid profiles" need clarification; such phrasing is ambiguous without context. The results are described without confidence intervals, p-values, or other statistical measures, which are essential to substantiate claims.

Response: Thank you for this important feedback that our abstract was not sufficiently succinct. We have modified the abstract according to your suggestions and the recommendation of reviewer No.4. We believe that the abstract, which is a compromise between reviewer comments and the space limitation, is stronger now.

Introduction

p. 3, lines 61–66:

The introduction does not sufficiently highlight gaps in existing literature to justify the study’s necessity. It assumes the audience is already aware of the limitations in current research on DMT1 and anticoagulation.

Response: Thank you for pointing out this gap. This part was indeed not emphasized, we have modified it to underline the need for such a study.

p. 3, lines 66–70:

Statements regarding the suitability of direct anticoagulants for DMT1 patients are broad and unsupported by specific prior studies or references.

Response: Thank you for this comment, as the previous version of the article was not written clearly enough, as there are no clear previous indices that direct anticoagulants should be used in preference in the diabetic population over other patients (e.g., replacement arthroplasty). We have modified that part, searched for available literature, and added clinically relevant data, and also one relevant experimental paper.

p. 3, lines 78–90:

The description of coagulation assays lacks precision. Specific references to how these tests uniquely contribute to the study goals could strengthen the argument.

Response: Thank you for this comment, which was also similar the reviewer No.3. It was quite unexpected that, notwithstanding these tests are frequently used, their relationship to cardiovascular morbidity and mortality has not been extensively reported. We found two relevant papers, which are mentioned in the introduction.

Methods

p. 5, lines 99–102:

No rationale is provided for choosing 50 participants per group. Was a power calculation conducted to ensure adequate sample size?

Response: Yes, we have estimated sample size: The number of patients (50 in each group) is sufficient to prove a difference of 10% in anticoagulant response (power analysis: p=0.05, power 0.95 → 26 patients in each group are needed). There is an additional important aspect. These results are part of a larger project where we screened the differences between different patients suffering metabolic diseases; hence, in order to make such a project feasible, we also calculated the number of patients also with regard to funding limitations. We have added information about power analysis in the statistical section of the paper.

p. 5, lines 101–105:

The criteria are described but lack justification. For instance, why were individuals with hypothyroidism or hypertension included if these conditions could influence coagulation?

Response: Thank you for this important comment. In principle, it is not easy to have a healthy control group encompassing a population with a large age range from 20 up to 60 years, where all donors will be fully healthy. Therefore, we had to accept some conditions, such as well-treated hypothyroidism and hypertension. Given the relatively frequent incidence of these conditions in the general Czech population, we considered it acceptable for inclusion. We have also added this information in the revised version of the paper.

Control Selection:

The control group, described as “generally healthy,” is not defined rigorously. Are they comparable to DMT1 patients in lifestyle and diet?

Response: This is an important point. In principle, diabetic persons should follow some recommendations about lifestyle and diet. It might not be easy to monitor them. The common food questionnaires are based on patient self-reporting and patients might not be fully sincere. In our „generally healthy“ population, we did not require specific diet or lifestyle modification. Our aim was to include a real population in real-life conditions. Anyway, we have stored some plasma at -80°C in which we will measure metabolites of (food) polyphenols. Unfortunately, the method for their detection is very ambitious and our colleagues from the Analytical department have not yet fully validated the method. So we cannot report these data at the moment. We have, however measured tryptophan in the serum, which might give us some outlook if there are differences. Indeed, there was a difference; serum level of tryptophan was 66.62 ± 10.08 μM in healthy donors and 58.69 ± 13.09 μM in DMT-1 patients (both data had Gaussian distribution, t-test analysis, the difference was significant at p=0.001). In addition, we measured some inflammatory markers. See, please, our new version of the paper, where we included these data. These markers are not specific to lifestyle but might give another outlook as they do not seem to be directly linked to other risk factors like LDL-cholesterol (our new Table 3).

Confounding Variables:

The study fails to address the influence of potential confounders such as inflammatory markers, diet, or medication use.

Response: As mentioned above, we measured some relatively newer inflammatory markers with a well-documented cardiovascular relevance (neopterin and the ratio kynurenine to tryptophan). We thought to keep these data for another publication, but based on the reviewer's comment, we added the data in the publication. We are not sure if the inflammation is a confounder, based on the aim of this study, we rather think that this is another risk factor of cardiovascular disease. Concerning diet, we can perform some analysis later when we have all the data. We apologize, but we cannot assess the relationship now. We mentioned it as a limitation of our study. Medication can definitely represent a confounder. We looked at drugs which were unequally distributed between generally healthy donors and diabetic patients (Table S1: ACE inhibitors, angiotensin II antagonists, HMG CoA reductase inhibitors, gabapentinoids and antihistamines) with exception of insulin which was logically given to all patients but not to healthy person and hence its impact cannot be drawn from our study. A new table (S2 Table in the supplementary file) based on these results was prepared. As the results were almost similar (there were no differences in significance), we suppose that these drugs have not influenced our results.

References:

Avanzas P, Arroyo-Espliguero R, Kaski JC. Neopterin and cardiovascular disease: growing evidence for a role in patient risk stratification. Clin Chem. 2009;55(6):1056-7. https://pubmed.ncbi.nlm.nih.gov/19395434/

Baumgartner R, Forteza MJ, Ketelhuth DFJ. The interplay between cytokines and the Kynurenine pathway in inflammation and atherosclerosis. Cytokine. 2019;122:154148.

https://pubmed.ncbi.nlm.nih.gov/28899580/

Teunis CJ, Stroes ESG, Boekholdt SM, Wareham NJ, Murphy AJ, Nieuwdorp M, Hazen SL, Hanssen NMJ. Tryptophan metabolites and incident cardiovascular disease: The EPIC-Norfolk prospective population study. Atherosclerosis. 2023 Dec;387:117344 https://pubmed.ncbi.nlm.nih.gov/37945449/

Wang Q, Liu D, Song P, Zou MH. Tryptophan-kynurenine pathway is dysregulated in inflammation, and immune activation. Front Biosci (Landmark Ed). 2015;20(7):1116-43. https://pubmed.ncbi.nlm.nih.gov/25961549/

Results

p. 8–12:

The narrative description of results omits some key statistical details. For example, confidence intervals for effect sizes are not reported.

Response: Thank you for pointing this out. Instead of adding key statistical details directly in the text, we have supplemented the missing information directly in the tables and graphs. We think that this is a better solution as the interested reader can find them in the paper, but many details in the text might render the part of the results less readable.

The table (1) with results was not completely clear in terms of statistical analyzes. We re-checked all data; in most cases, the data did not have a Gaussian distribution, so they are reported currently as median and 95% confidence intervals. In case of the Gaussian distribution, we reported the data as mean ± SD.

p. 9–12:

There is minimal discussion of non-significant findings. This could introduce bias in interpreting the data.

Response: We understand this point and we agree that some non-significant results should also be discussed. We modified the discussion by the addition of other points based on your and other reviewer comments. Logically, due to the fact that our paper produced a large dataset, not all insignificant findings could be discussed. Current discussion already has 7 pages so we think that additional prolongation will rather be counterproductive.

Ambiguity in Causality:

Correlations between vitamin K levels and coagulation outcomes are mentioned, but causation is implied without robust evidence.

Response: This is also a very good point. As far as we know, we are the first to measure different forms of vitamin K in patients and to correlate them also with coagulation. The physiology of vitamin K is very complicated (e.g. our recent paper on vitamin K in Nutr Rev or even very fresh papers on vitamin K in relation to inflammation, Lacombe et al., Xie et al.). Even if we could not bring some conclusive data at the moment, we have discussed this issue in more detail. Also, the addition of inflammatory markers to this paper increased the informative value of this paper.

Reference:

Lacombe J, Ferron M. Vitamin K-dependent carboxylation in β-cells and diabetes. Trends Endocrinol Metab. 2024;35(7):661-73 https://pubmed.ncbi.nlm.nih.gov/38429160/

Mladěnka P, Macáková K, Kujovská Krčmová L, Javorská L, Mrštná K, Carazo A, Protti M, Remião F, Nováková L; OEMONOM researchers and collaborators. Vitamin K - sources, physiological role, kinetics, deficiency, detection, therapeutic use, and toxicity. Nutr Rev. 2022;80(4):677-698. https://pubmed.ncbi.nlm.nih.gov/34472618/

Xie Y, Li S, Wu D, Wang Y, Chen J, Duan L, et al. Vitamin K: Infection, Inflammation, and Auto-Immunity. J Inflamm Res. 2024;17:1147-60 https://pubmed.ncbi.nlm.nih.gov/38406326/

Demographic Stratification:

While stratified analyses by age and BMI are conducted, their clinical implications are not clearly discussed.

Response: Similarly to a previous comment by the reviewer, we find it very relevant. It is not very simple to discuss these data with a relatively limited number of patients in particular with higher age. Regardless, as we included inflammatory markers, they helped us to understand at least partly the relationship. We discussed these aspects at least in part in our new discussion.

Discussion

p. 14–17:

Claims regarding the protective effects of lower cholesterol and vitamin K levels in DMT1 lack robust evidence. The discussion overstates the findings’ implications.

Confounding Factors:

The potential influence of unmeasured variables (e.g., inflammatory states or micronutrient deficiencies) is not adequately acknowledged.

Response: The reviewer is correct that our data cannot be fully conclusive given the limited number of patients and a limited number of tested parameters. Concerning inflammation, this can definitely affect blood hemostasis. We have added new data which shows a relationship between inflammation and responses to the most active direct anticoagulants. From micronutrients, we assessed solely vitamin K as it has a direct role in blood coagulation. We were not able to test other micronutrients due to their high numbers. Regardless, we have included a limitation section in our paper, where we emphasized the possible dietary differences.

Comparison to Prior Studies (p. 14):

The discussion does not sufficiently critique or compare the study's results with contradictory findings from other research.

Response: We apologize, we tried our best to find relevant papers on the topic. In this improved version of our paper, also in relation to the comments of other reviewers, we have added more papers. If the reviewer still thinks that some specific additional paper(s) should be discussed, please let us know.

Limitations (p. 17):

Limitations are inadequately addressed. For example, no mention of the small sample size or lack of long-term follow-up is made.

Response: We agree with the need to better address the limitations of our paper. We have included this aspect in the new version of our manuscript.

p. 18:

The conclusion is overly broad and lacks actionable insights for clinical practice.

Response: Establishing a clear clinical outcome from our results is not simple, but we have carefully modified the conclusion section in relation to this aspect.

Reviewer #3: The authors performed an interesting study. There are potential issues in methodology and thus interpretation of results. Specifically, I have the following comments:

1. I am not sure whether PT/APTT are regularly used to assess the prothrombotic burden

Response: Both PT and aPT

---

## [Decision Letter · Decision Letter 1]

PONE-D-24-43881R1

Type 1 diabetes mellitus patients had lower total vitamin K levels and increased sensitivity to direct anticoagulants

PLOS ONE

Dear Dr. Přemysl Mladěnka ,

Thank you for your revised submission of the manuscript entitled ”Type 1 diabetes mellitus patients had lower total vitamin K levels and increased sensitivity to direct anticoagulants” to PLOS One. After careful consideration of the reviewers’ comments and your thorough responses, I am pleased to inform you that your manuscript has been accepted for publication.

While Reviewer #4 recommended rejection based on specific methodological concerns, I carefully reviewed both their critique and your detailed, point-by-point responses. In my judgment, these concerns were appropriately and comprehensively addressed through robust clarifications, relevant literature citations, and meaningful revisions throughout the manuscript. It is also important to note that your study does not seek immediate clinical application, but rather provides novel comparative data that contribute to hypothesis generation and lay groundwork for future research in this field. I would also like to take this opportunity to acknowledge and sincerely thank Reviewer #4 for their detailed and constructive critique, which—despite their final recommendation—contributed significantly to improving the manuscript’s clarity and scientific rigor.

Based on the overall merit of the revised manuscript, the scientific value of the findings, and your thoughtful engagement with the reviewers’ feedback, I have made the decision to accept your manuscript for publication in PLOS One.

Congratulations again on your acceptance, and thank you for choosing PLOS One as the venue for your work.

Best regards,

Yousef Khazaei Monfared 

Reviewers' comments:

Reviewer's Responses to Questions

**Comments to the Author**

1. If the authors have adequately addressed your comments raised in a previous round of review and you feel that this manuscript is now acceptable for publication, you may indicate that here to bypass the “Comments to the Author” section, enter your conflict of interest statement in the “Confidential to Editor” section, and submit your "Accept" recommendation.

Reviewer #1: All comments have been addressed

Reviewer #2: All comments have been addressed

Reviewer #3: All comments have been addressed

Reviewer #4: All comments have been addressed

Reviewer #5: All comments have been addressed

2. Is the manuscript technically sound, and do the data support the conclusions?

Reviewer #1: Yes

Reviewer #2: Yes

Reviewer #3: No

Reviewer #4: Yes

Reviewer #5: Partly

3. Has the statistical analysis been performed appropriately and rigorously? 

Reviewer #1: Yes

Reviewer #2: Yes

Reviewer #3: No

Reviewer #4: Yes

Reviewer #5: Yes

4. Have the authors made all data underlying the findings in their manuscript fully available?

Reviewer #1: Yes

Reviewer #2: Yes

Reviewer #3: No

Reviewer #4: Yes

Reviewer #5: Yes

5. Is the manuscript presented in an intelligible fashion and written in standard English?

Reviewer #1: Yes

Reviewer #2: Yes

Reviewer #3: Yes

Reviewer #4: Yes

Reviewer #5: Yes

6. Review Comments to the Author

Reviewer #1: Thank you. The paper is completed and all the comments have been addressed. After receiving the other reviewers responses,it's ready to publish.

Reviewer #2: I have no further comments. Thank you for addressing my comments carefully for a more sound manuscript.

Reviewer #3: The authors made an attempt to address the comments. However, the aims/objectives are still not clear. If the authors want to investigate the effect of different NOACs, then reporting APTT/PT is not enough. Although one may argue that there is correlation, still, measuring the NOAC levels and clinical event is necessary.

It is also unclear for the clinical implications. The authors reported that prolonged APTT/PT is associated with higher atherosclerotic risk. So taking NOACs will result in higher atherosclerotic risks? There are many patients with AF and CAD taking NOACs for prevention of cardiovascular event.

Reviewer #4: (No Response)

Reviewer #5: The revised manuscript has undergone significant improvements and is now suitable for publication.

7. PLOS authors have the option to publish the peer review history of their article (what does this mean? ). If published, this will include your full peer review and any attached files.

**Do you want your identity to be public for this peer review?** For information about this choice, including consent withdrawal, please see our Privacy Policy .

Reviewer #1: **Yes: ** Laleh Abadi marand

Reviewer #2: No

Reviewer #3: No

Reviewer #4: No

Reviewer #5: **Yes: ** Kandahalli Venkataranganayaka Abhilasha

---

## [Editor Report · Acceptance letter]

PONE-D-24-43881R1

PLOS ONE

Dear Dr. Mladěnka,

I'm pleased to inform you that your manuscript has been deemed suitable for publication in PLOS ONE. Congratulations! Your manuscript is now being handed over to our production team.

Kind regards,

on behalf of

Dr. Yousef Khazaei Monfared

Academic Editor

PLOS ONE